# Driving sustainability transitions through financial tipping points

Nadia Ameli, University College London, London, UK, n.ameli@ucl.ac.uk

Hugues Chenet, IESEG School of Management, Univ. Lille, CNRS, UMR 9221 – LEM – Lille Economie Management, F-59000 Lille, France ; and University College London, London, UK, h.chenet@ieseg.fr

Max Falkenberg, City University of London, London, UK, max.falkenberg@city.ac.uk

Sumit Kothari, University College London, London, UK, sumit.kothari.16@ucl.ac.uk

Jamie Rickman, University College London, London, UK, jamie.rickman@ucl.ac.uk

Francesco Lamperti, Scuola Superiore Sant'Anna, Pisa, Italy, francesco.lamperti@santannapisa.it

## Abstract

Achieving a net-zero carbon economy requires significant structural changes in the financial system, driving a substantial shift in investment towards low-carbon assets. This transformation of finance is necessary beyond the objective of climate stabilisation, but is more broadly required to foster sustainably thriving economies. In this paper, we offer a critical discussion of the positive tipping points that can be activated in the financial system to drive a fast, sustainable transition. Indeed, the identification and activation of critical and positive tipping points can lead to the amplification of sustainable investments and foster transformative changes in the practices of the financial sector. Through the alignment of expectations, steering of herding behaviour, mobilisation of public finance, reduction of capital costs, attainment of low-carbon investment thresholds in developing nations, and implementation of robust financial regulations and policies, the financial system can assume a central role in re-orienting economies onto a net-zero and sustainable course. Taken together, such mechanisms highlight the positive tipping points that can be triggered within sustainable finance and emphasise the necessity of policy interventions to activate and capitalise on these dynamics.

## Keywords

Sustainable finance, tipping point, expectations alignment, feedback loop, path-dependency, investment threshold and dynamics, herding behaviour, financial regulation and policies

## 1. Introduction

Scientific consensus regarding the need to reduce increasing resource demands is unequivocal (IPCC 2022, Richardson et al 2023), as humanity faces a confluence of urgent environmental challenges, including climate destabilisation and biodiversity collapse. In the face of this unprecedented situation, the financial system is called upon to play its part in shifting the economy back towards a 'safe operating space' (Rockström et al 2009). This necessitates a rapid shift from financing the 'undesirable' (i.e. the 'dirty', the 'harmful') to financing the 'desirable' (i.e. the 'sustainable', the 'green'). However, the primary function of the financial system, as widely accepted within most advanced (market-)economies, is to maximise financial risk-adjusted returns. Finance is agnostic with respect to the greenhouse gas emissions of its activities, or whether they promote or harm biodiversity. The effective implementation of finance in a sustainable manner, or 'sustainable finance', is thus not assured. Indeed, the current economic paradigm to which finance predominantly adheres is based on ever-rising demand, short-term profitability, inadequate environmental policy and unclear industrial priorities at both national and international levels. In this context, perpetuating historical patterns remains the preferred approach for the financial sector to ensure profitability, and as such it fails to fulfil its transformative role (Ameli et al. 2019, Christophers 2022). Driven by backward-looking, climate- and nature-blind indicators, and ignoring the complexity and systemic impacts of their investments on the environment (Chenet et al. 2021, Crona et al. 2021), financial actors continue to allocate massive amounts of capital to environmentally damaging industries, such as fossil fuel assets and deforestation. This practice consolidates carbon lock-ins and contributes to long-term biodiversity decline (FTM 2023, Ruzzenenti et al 2023, Svartzman et al 2021, Kedward and Ryan-Collins 2022). Ironically, by doing so, the financial sector is driving the accumulation of environment-related financial risks to which, by its own admission, it is now dangerously exposed (Chenet 2024).

Our paper discusses a number of mechanisms that may push the financial system towards positive tipping points, potentially triggering transformative change across the real economy by influencing the volume of financial flows and the associated costs. Tipping points describe critical thresholds in a complex system that, if crossed, activate self-perpetuating processes of change that drive the system into a qualitatively different state (Lenton 2020). Here, the complex system under examination is the financial system, broadly defined as the set of banking and non-banking financial institutions, regulatory bodies and investors, along with the market and non-market relationships they share among themselves and with the real economy. Especially after the Global Financial Crisis (2008-2009), the financial system has been increasingly understood as a complex system (e.g. Farmer and Foley 2009; Dosi and Roventini, 2019), that is, a system composed of heterogeneous interacting entities characterised by varied emergent properties at the macro level which are shaped by the structure and dynamics of these interactions. The architecture of the financial system determines the direction and allocation of financial flows to different economic actors and sectors, thus propelling activities in favoured segments of the economy with substantial financial capital, while constraining activities in less favoured areas. Governments, central banks and regulatory authorities through the exercise of their powers to frame policy and regulations, can alter the structure and the dynamics of the financial system. This provides the opportunity to activate positive tipping points leading to a structural transformation of the real economy.

Here, we focus our analysis on positive tipping points, which describe how social, political,
economic or technological systems can shift rapidly into new system states (Tabara et al.
2018), that are less harmful, or even offer solutions to the challenge of climate change. While
the examples discussed herein predominantly focus on climate finance, similar reasoning and
principles can be applied to broader sustainability issues, such as biodiversity. Indeed, the
financial sector is currently modelling its approach to biodiversity finance on climate finance
principles developed over the past decade (Chenet 2023). Transformation of the financial
system is not the singular, definitive solution capable of addressing all environmental
challenges. Finance functions merely as a tool, affecting change through its interactions with
the real economy, and should be viewed as part of a broader strategy incorporating, for
example, industrial policy, transition planning, social justice, and changes in consumption
habits. This holistic approach is crucial to ensure a long-term equilibrium of humanity within
planetary boundaries. Our objective is to leverage the theoretical and empirical aspects of the
financial system, as it is or as it could be reimagined, to explore how it could more effectively
address the systemic challenges we are facing. Rather than presenting a prescriptive solution,
our efforts represent an initial inventory of potential tools. We thus try to provide a broad
overview of how tipping points may facilitate the transition to sustainable finance, while
recognising the composite nature of the financial system. Some dynamics may hold relevance
across diverse contexts globally, others are more suitable for specific sectors, regions or
stakeholders.
The next sections are organised as follows. Section 2 discusses the role of the financial system
with respect to the problem of sustainability and climate change in particular; section 3
provides a critical overview of the positive tipping points that may be activated in the financial
system and offers a (non-exhaustive) review of the available empirical and modelling
evidence; finally, section 4 concludes the paper and summarises the key points.
**2.  The financial system in the face of environmental challenges**
In the 2000s, the financial sector was largely absent from the key discussions on climate
change and the environment. Banks' action on climate change was limited to reporting on the
efficiency of their light bulbs and reducing business trips, with no mention of the detrimental
consequences of their increasing lending to fossil fuels companies.[1] An important milestone
was the 2015 Paris Agreement, which explicitly acknowledged the role of finance in
addressing climate change through Article 2.1(c) (Zamarioli et al. 2021). Although its full
implementation is still pending, it triggered a new institutional regime and narrative related to
finance and climate change, highlighting the responsibility of the financial sector to shift the
economic pathway in line with climate targets. In the same year, Mark Carney's speech on
financial stability and the risks associated with climate change (Carney, 2015) brought the
topic of climate-related financial risk to the fore. By emphasising the urgent need for financial
institutions to adopt climate risk management and reporting measures 'before it's too late',
Carney initiated an important climate move, mainstreaming climate change in discussions of
the financial sector's practices and regulations. Fully establishing transparency across the
financial system thereby became a prime goal of financial policy, regulation and industry

---

[1] See e.g. BNP Paribas Annual Report 2005 - https://invest.bnpparibas/en/document/annual-report-2005 [pp. 68-72, 107-113, 330-344]

efforts in the climate finance arena (Ameli et al 2021a). A similar path was recently followed
by financial institutions and authorities concerning biodiversity (Chenet 2023, 2024). In some
respects, Carney's speech can be seen  as an institutional tipping point for sustainable finance
that kick-started discussions, voluntary initiatives and, eventually, regulatory mandates that
have led to distinct changes in the financial sector's operations and practices.
More recently, the establishment of initiatives such as the private sector-led Glasgow Financial
Alliance for Net Zero (GFANZ) and the central banks-led Network for Greening the Financial
System (NGFS), have demonstrated the growing commitment of financial institutions, from
commercial entities to public authorities, to align themselves with climate targets. GFANZ
signatories committed to reaching net-zero carbon emissions by 2050, in a manner that is in
line with the +1.5°C target (i.e., with limited temperature overshoot and using existing
technologies). This marked the first instance in which financial institutions committed and
pledged to align with climate targets.[2] On the financial authorities side, the NGFS created a
landmark governance framework to better coordinate and regulate financial institutions in
addressing climate change. Given their status and regulatory strength within the financial
system, this has provided a strong signal to financial institutions worldwide that a low-carbon
transformation of their activities is imminently needed.
This sequence of events can be viewed as the initial catalyst, or accelerator (cf. GTPR2023,
Fig.2 p.33), for challenging traditional practices in the financial system, prompting financial
actors to embark on a different path in terms of their investment outlays (Farmer et al. 2019).
These initial shifts have the potential to cross critical thresholds (i.e. 'tipping points'), where a
minor alteration can trigger a larger and systemic change, and where nonlinear feedback
effects act as amplifiers of such change (Lenton et al. 2022). By influencing the allocation of
capital to different sectors or activities, the financial system has indeed the power to affect the
evolution and composition of the real economy, thereby opening the way to the emergence of
tipping points across sectors.
In a variety of historical episodes, the financial system has acted as an amplifier of shocks,
both positive and negative. This phenomenon is commonly referred to as the 'financial
accelerator' (Bernanke et al. 1999; Delli Gatti et al. 2010), which describes how developments
in financial markets amplify and propagate the effects of minor changes in the economy. For
example, bursts of financial bubbles have triggered uncertainty, instability, contagion among
financial actors, and feedback loops that cause ripple effects in the real economy, even though
the initial shock was not particularly severe. The Global Financial Crisis of 2008 is a prominent
example of such a negative shock. On the other hand, financial accelerators have the potential
to amplify positive shocks through, for example, mechanisms which dampen the financial
fragility of firms operating in the real economy or enhance the effects of innovation and its
diffusion, resulting in positive outcomes in the medium and long run (Lamperti et al. 2021).
Similarly, favourable financial conditions can magnify the impact of policies aimed at
sustaining aggregate demand, creating significant synergies between prudential, fiscal, and
monetary measures.

---

[2] NB: the efficiency of these initiatives is nevertheless questioned, from the business-as-usual of financing decisions (e.g. Sastry et al 2024) to the current 'ESG backlash' in the US (e.g.'The real impact of the ESG backlash', FT 2024, https://www.ft.com/content/a76c7feb-7fa5-43d6-8e20-b4e4967991e7, 'Insurance industry turmoil over climate alliance exodus', FT 2023, https://www.ft.com/content/1dd66ce1-a720-4c56-96d9-8d47f07f376f ).

Finance can also have a more direct impact on the real economy. Following Perez (2003),
financial actors and, more prominently, public investors (Mazzucato 2013) play a central role
in enabling technological revolutions by actively contributing to the advancement and
implementation of innovative processes, technologies and services, extending their
involvement beyond simply providing funds. In fact, they often take part in the management of
the innovation process, assuming the role of financial entrepreneurs and 'picking winners'. But
other mechanisms can also operate concurrently. For instance, once a particular path is
established, financial behaviours can lead to a self-reinforcing cycle where an accepted choice
gains momentum and becomes increasingly difficult to change (Arthur, 1989). Also, financial
markets have a tendency to replicate the economy as it is and resist making potentially costly
new decisions. Finance thus has the capacity to both expedite or impede the dissemination of
new products and technologies, particularly those of utmost importance for the transition to a
low-carbon future.

## 180 3. Finance and positive tipping points

### 181 3.1 The potential for positive tipping points in sustainable finance
In this section, we outline and critically discuss mechanisms that exhibit the potential to
leverage tipping points in the financial system, with a particular reference to investments
towards low-carbon assets and technologies.
Theoretical and empirical evidence suggests that public finance has a catalytic role for
mobilising investments (Mazzucato 2013). Indeed, the ability of public actors (e.g. public
investment banks, governmental agencies) to take on risk induces private investors to follow.
This is not only due to the substantial amount of funding provided by public actors, but also
because of the quality of financing they offer. Public financing, with its long-term horizons,
favourable repayment conditions and ancillary support, resembles the role of financial
entrepreneurs. By underwriting risks associated with low-carbon investments and supporting
specific technological trajectories using green subsidies, public finance can mitigate market
uncertainty, potentially creating tipping points in the financing of low-carbon projects and
assets (Campiglio and Lamperti 2021; Mazzucato and Semieniuk 2018). However, the
emergence of positive tipping points cannot be easily guaranteed and needs adequate policy
support. For example, a mission-oriented industrial policy shaping the behaviour of financial
actors under direct or indirect public control (e.g. public investment banks, development banks,
government agencies, large public utilities) can increase the likelihood of positive tipping
points in the dynamics of investments and, hence, aggregate production (Dosi et al. 2023).
Expectation alignment on the timing and speed of the transition may also act as a tipping point
with the potential to significantly scale up sustainable investment (Campiglio and Lamperti
2021; Campiglio et al. 2023). Uncertainty about the future prospects of low carbon assets
coupled with unclear information about the strength of climate policy may delay substantial
portfolio rebalancing decisions. In such cases, investors may adopt a more cautious 'wait-and-
see' approach, favouring conventional investments whose profitability appears less affected
by unclear climate policies. On the contrary, certainty regarding future climate policy schedules
through legally-binding climate commitments, carbon budgets and strategic plans, can signal
the long-term trajectory of the economy, inducing a positive correlation between low-carbon
asset returns and macroeconomic performance. This alignment of beliefs can coordinate and
shift the strategies of long-term institutional investors (e.g. pension funds), which are typically
influenced by a wide range of subjective beliefs about asset returns (Broeders and Jansen,
2021). Hence, aligning expectations on the timing and speed of the low-carbon transition could
mitigate risk and spur momentum towards sustainable investments. A shift in the investment
behaviour of large financial actors may push the financial system past a tipping point resulting
in a self-reinforcing cycle in which sustainable investments become increasingly attractive,
transforming from mere diversification assets into strategic ones (see Figure 1).

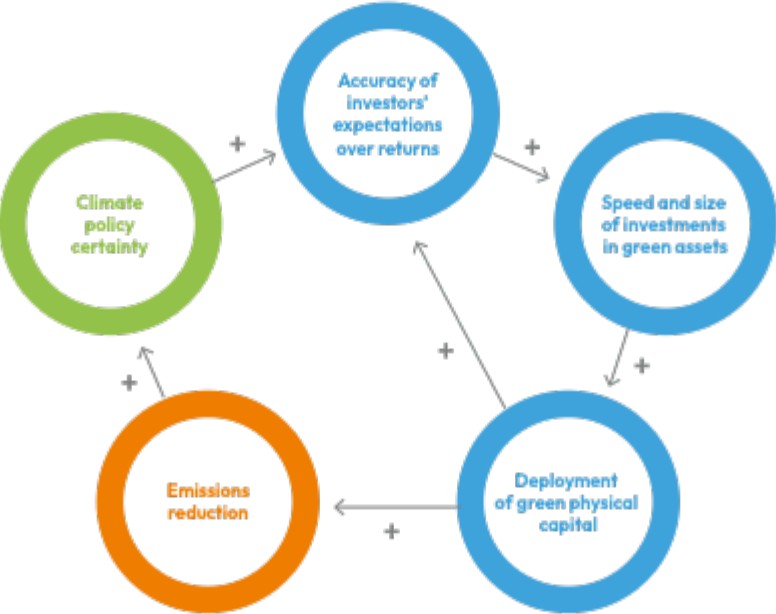


**Figure 1:** *A positive feedback loop favouring a tipping point in the dynamics of low-carbon*
*investments. The set of self-reinforcing mechanisms and feedback loops occurring in the process*
*between climate policy certainty and deployment of green physical capital. Expectation alignment*
*creates a positive feedback which can be triggered and sustained by certainty in climate policy. The '+'*
*symbol indicates a positive effect (Source: Ameli et al. 2023).*

Tipping points in financial markets can also emerge through herding behaviour, wherein a
critical mass adopts a particular trend, ultimately influencing the broader population to follow
suit (Bikhchandani and Sharma, 2000). Herding behaviour refers to the tendency of investors
to mimic others, especially during periods of uncertainty or when faced with limited information,
resulting in the amplification of market movements. In the context of financial tipping points,
herding behaviour can have both positive and negative impacts. On the one hand, it can
exacerbate market instability and contribute to the formation of speculative bubbles. When
investors flock towards certain assets or sectors, it may lead to an unsustainable surge in
prices and valuations. However, on the other hand, herding behaviour can also be channelled
positively to drive sustainable investments and foster the transition towards a low-carbon
economy. For instance, policy action targeted at the global systemically important banks (G-
SIB) to ensure financial stability by better managing transition risks in their portfolios, can
induce sector-wide portfolio rebalancing away from fossil fuel investments that are mis-aligned
with climate goals and carry stranded asset risks (International Monetary Fund 2023).
Similarly, critical mass comes into play when a sufficient number of investors adopt
sustainable practices (e.g. if GFANZ were to become a dominant approach) or allocate funds
to sustainable investments. This creates a self-reinforcing cycle, attracting more capital and
generating increased demand for sustainable products and services. Of course, such a self-
reinforcing mechanism should by no means give rise to a 'green bubble'. The significance of
herding and critical mass lies in their potential to facilitate the scaling up of sustainable
investments. Herding behaviour can rapidly accelerate the adoption of sustainable
investments until a critical mass is reached. Once this tipping point is reached, it becomes
easier for sustainable investments to attract more funding and support from a widening pool
of investors. This positive feedback loop can lead to a transformative shift in the financial
landscape, where sustainability becomes the new norm rather than the exception.
These individual tipping points in financial markets signal the existence of sensitive
intervention points (SIPs), i.e. identifiable opportunities for deliberate actions that can trigger
associated tipping points. SIPs can either be small 'kicks' that trigger positive feedback cycles
in a system, or can drive a systemic shift in the inherent dynamics of a system that lead to
transformative changes even without external triggers (Sharpe and Lenton 2021; Farmer et
al. 2019). Activating an SIP initiates tipping dynamics, causing significant shifts in the market.
Policy intervention can serve as the catalyst for such changes directly, by providing the initial
'kick', or indirectly, by shifting the underlying dynamics that bring about the transformation.
Additionally, Farmer at al. (2019) identified two finance-related SIPs. The first involves
financial disclosure and falls into the 'kick' type of SIP. Indeed, a change in accounting
standards and disclosure guidelines to measure and report climate-related financial risks
complemented by policy initiatives such as green taxonomies and sectoral transition plans,
could trigger a substantial repricing of fossil assets, such as fossil fuel reserves and securities
valuations. Consequently, this would limit the ability of the fossil fuel sector to invest in new
fields, thereby reducing committed emissions. Preventing such investments lowers the
economic, social, and political costs of transforming the energy industry, as it levels the playing
field for renewables, reduces the risk of stranded assets, and enhances the credibility of
climate targets. Here, the mechanism at stake relies on market efficiency, a theory where
information availability is core to investment decisions and its relevance, in terms of optimal
capital allocation. Based on this disclosed information, risk/return expectation will be the prime
– if not the sole – guide for financial institutions, which would then contribute to the transition
with no need to have any extraneous intention to align their portfolio with such transition goals.
Disclosure of environment-related financial risk has been the most prominent mechanism
promoted by financial authorities and institutions over the last decade, despite its inherent
limits (Ameli et al 2020, 2021b). These concern the extent to which markets can effectively
incorporate disclosed financial risk information in asset prices without any long-term guidance
concerning an inherently uncertain and evolving low-carbon transition. The progressively more
'interventionist' regulatory propositions, especially in Europe, can be seen as attempts to
correct these limits.
The second SIP pertains to technology selection and a targeted 'shift' towards low-carbon
investment. Contrary to traditional portfolio theory, diversification of investments can be
detrimental, especially when it comes to developing novel and uncertain technologies where
spreading resources too thin can hinder significant progress. Instead, rapid progress requires

concentrating resources on specific technologies (Way et al. 2019). For example, solar PV has achieved remarkable progress due to targeted support, becoming cheaper than most alternatives. The next step is to similarly focus on developing technologies that can accelerate the deployment of solar PV, such as energy storage. In essence, inducing a tipping point in this context involves not attempting to invest across a broad range of options with hopes of developing each of them but concentrating efforts on technological complementarities that synergistically support research, development, and actual deployment. Further, identifying these technological complementarities dramatically reduces technological uncertainty, which would amplify the dynamics of technology diffusion even further. In contrast to pure market mechanisms, such choices may be directly or indirectly fostered by public sector interventions, in line with some sustainability transition planning. The objective here is to align financial portfolios with an environmental goal or scenario.

There may however be trade-offs involved between the two SIPs wherein the policies and practices related to disclosure of climate-related financial risks and portfolio realignment may result in lower investments in low-carbon projects due to a higher perception of transition risks. This is possible both in cases of 'bridge' technologies that may have uncertain prospects in the longer term, such as hydrogen-fuelled transport or storage solutions, and innovative low-carbon technologies, such as marine power, whose future cost and deployment trends are highly uncertain. The inherent uncertainty of the energy transition may create higher perception of risks due to indeterminate eventual outcomes, specific technological trajectories or timing of different climate-mitigating actions. Strong policy choices, however, can foster market confidence, despite risks of inefficiency, to create a conducive  environment where portfolio realignment is accompanied by higher investment in technologies necessary for a timely energy transition.

**3.2 Empirical and modelling evidence of tipping points in sustainable finance**

In terms of empirical and modelling evidence, a variety of examples show how the financial system can play a pivotal role in activating tipping points to accelerate the transition to a net-zero carbon economy.

In developing countries, policy support can help to overcome climate investment traps created by the high costs of accessing finance (Ameli et al 2021a). Access to finance, understood as the costs of raising funding for a specific project from different sources, varies significantly across countries. For instance, in some African nations, such as the Democratic Republic of the Congo, Madagascar and Zimbabwe, the cost of capital can soar to 30%, while in developed countries such as Germany and Japan, it can be as low as 3% (Ameli et al 2021a). The high cost of accessing capital is preventing developing countries from decarbonizing their economies. Levelling the finance playing field could thus help poorer nations to steer their economies onto a net-zero course.

While energy system transitions in developing economies require particularly high investment, these parts of the world are also particularly financially constrained. They are characterised domestically by under-developed capital markets and lack of capital stock (Ameli et al 2021a). Furthermore, international finance is restricted due to high sovereign and local currency risks. Projects funded with foreign currency while generating returns in local currencies lead to

volatile economic fundamentals (Ameli et al 2021b, Bilir et al 2019), resulting in restricted
access to external funding sources. This leads to a chronic lack of available finance to support
low-carbon investments, creating a climate investment trap which occurs when climate-related
investments remain chronically insufficient, with dynamics similar to those of the poverty trap
(Ameli et al 2021a). A self-reinforcing cycle takes place where high risk perceptions lead to
increased capital costs, delaying the transition to cleaner energy systems and carbon emission
reductions. Climate change impacts exacerbate the situation (IPCC 2022), causing adverse
impacts on production systems, economic output, unemployment, and political stability (see
Figure 2).
Policies that reduce capital costs, such as credit guarantee schemes, foreign exchange
hedges and political risk insurance can shift risk away from private investors resulting in a
lower cost of capital that may act as a tipping point for low-carbon technology deployment and
allow developing economies to achieve large sustainable energy capacity and faster
emissions reduction. In the case of Africa, reducing the cost of capital by 2050 would allow
the continent to reach net-zero emissions approximately 10 years earlier than when reduction
is not considered (Ameli et al, 2021).

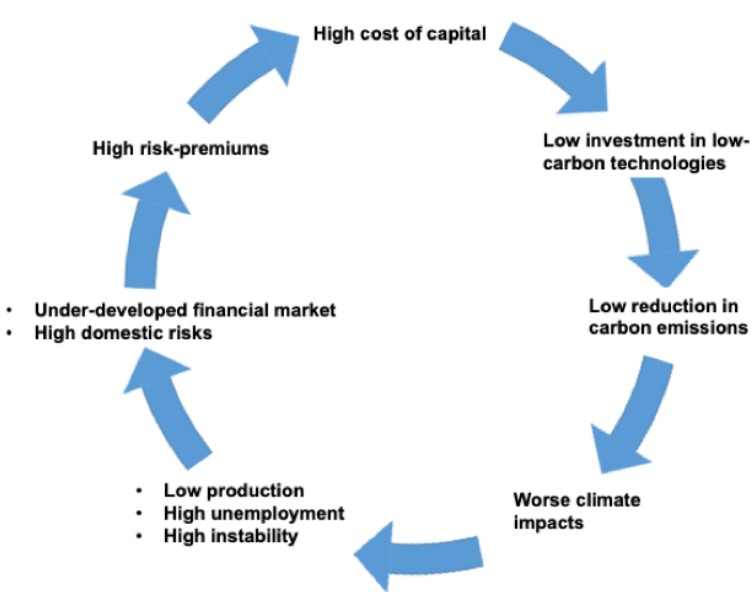


**Figure 2:** *A climate investment trap. The figure shows the set of self-reinforcing mechanisms and*
*related links occurring in developing economies characterised by high cost of capital. The strength of*
*these links is strongly linked to local conditions implying that the set of self-reinforcing mechanisms*
*could be exacerbated (or less relevant) in some economies. Note that some mechanisms are more*
*relevant at global/regional levels through aggregations across developing countries. For instance, local*
*carbon emissions are not necessarily linked with local climate impacts.*

Additionally, the flow of international capital into renewable projects in developing countries is
influenced by path-dependency, creating a tipping point in the scaling up of renewable
investments (Rickman et al. 2023). Countries with a track record of renewable investments
are more likely to attract future investments leading to positive feedback loops within
renewable energy markets. As countries build a track record in renewables, market confidence
grows, bringing down financing costs and attracting further investments in a virtuous cycle
(Egli et al 2018). Climate investment thus evolves through the strengthening of historical
investment and capital stock, rather than new investment. However, this also results in an
'investment lock-in' across countries as well as income groups, with only a small fraction of
countries receiving the majority of investment. Between 2010 and 2019, 76% of private capital
and 67% of public funds went to the top eight recipient countries (Rickman et al. 2023).
Evidence of path-dependency thus implies a new mechanism of the 'climate investment trap'
whereby historical inequities in financing are locked-in across countries and income groups
and perpetuate over time. To escape this investment lock-in, developing countries must
mobilise sustained investment to build a renewables track record that creates market
confidence and attracts private finance. Indeed, there is a non-linear relationship between the
probability of private investment in developing countries and their track record in renewables
investment, as measured by installed capacity. Once a significant capacity base of around
1GW (of wind or solar) is installed a tipping point is reached and the probability of private
investment increases sharply (Rickman et al. 2023). Crucially, low-income countries (e.g. in
Sub-Saharan Africa) fall far below this threshold, highlighting the inefficiency of opening
finance channels into poorer nations without sustained investment which can mobilise private
finance at scale. Investment decisions by public actors should thus move beyond project-
specific inducements to support more holistic renewable roadmaps and unlock developmental
co-benefits (Schwerhoff and Sy, 2017). Innovative financial and policy mechanisms, such as
transition plans with public funding from multilateral agencies and associated labelled
transition financing products, can target the evolution of the sector and build networks of
relationships in the financial sector to initiate path-dependent flows from private sources (Ameli
et al 2021b) and leverage tipping points in the renewable finance ecosystem.
Inducement effects between investors are another example of tipping points that can be
leveraged in sustainable finance. Financing in renewables markets is driven by a
heterogeneous set of actors spanning energy, financial, utilities and diversified sectors
(Mazzucato and Semieniuk 2018), who invest according to their investment remits,
preferences and capacities, as well as technological maturity and the market environment.
They collaborate across the development and operational stages of a project based on their
risk appetite and expected return, contributing different types of capital to the project in the
form of equity and loan investments. Their interaction and relationships drive the market
growth and technological maturity of renewable technologies within the energy system
resulting in unique emergent characteristics of the renewables sector across countries based
on their enabling investment environments.
In solar finance markets, co-investment relationships between different actors are established
at different stages of the market's development and evolve with the continued growth of the
sector (Kothari et al 2024). Actors exercise influence over their peers by inducing them into the
market and leveraging their investments alongside their own. These facets of relationships
differ between different actors in the solar sector based on existing co-investments, market
position of actors and the alignment of their interests. As markets evolve and different actors
enter the market, these processes of influence create tipping points in investment trajectories.
In the initial stages of the market development, for instance, investments by government
agencies induce investment by international institutions, supporting the initial deployment of
the technology. Increasing investments by renewable energy companies similarly influences
the actions of state-owned and private utilities. As markets grow, the involvement of
institutional investors creates the largest leverage (i.e. the amount of investment attracted)
through sizable investments from the private banking sector who are their natural debt partners
in renewable projects. The development of this relationship thus creates large flows of
investment into solar energy as a result of their investment dynamics.
Country context also determines the structure of solar finance markets and the strength of
relationships between different actors. The influencing power of different actors differs
significantly across countries. For example, in the United States, private bank lending induces
investments from a range of energy and diversified sectors, whereas in China, government
agencies and state-owned banks are major influencers and in Germany, renewable energy
companies and state-owned utilities exert a strong influence (Kothari et al 2024). From a policy
standpoint, therefore, it is important to consider the impact of policy instruments on prominent
actors in solar financing and the relationships that are driving the markets. Creating incentives
for these actors or using the relationships formed by government agencies and state-owned
actors effectively, can induce other actors into the markets and trigger non-linear growth of
investment, particularly from the private sector.
Theoretical modelling also reveals tipping points in the global network of banks which supply
debt to the fossil fuel industry (Rickman et al. 2024). A sharp decline in fossil fuel use is
necessary to achieve the Paris Agreement target of keeping global temperature rise below
1.5°C (Tong et al. 2019) and this will require a corresponding decline in bank lending to the
fossil fuel sector (Kirsch et al. 2021). However, mainstream financial theory holds that debt
flows to the fossil fuel sector will be resilient to the phase-out of lending by climate-friendly
banks, as their capital can simply be substituted by banks with a neutral stance on the climate
transition (Ansar et al. 2013). Capital substitution thus poses a challenge to a system-wide
decline in fossil fuel lending in an unregulated market. Macroprudential tools[3], such as capital
requirements rules, can counteract capital substitution by disincentivizing, or setting a limit, on
the amount of fossil fuel assets a banks' can hold, depending on their capital reserves. Models
suggest that while fossil fuel debt markets are resilient to the unregulated phase-out of capital,
the introduction of carbon-tilted macroprudential regulation can trigger a rapid contraction of
fossil fuel debt flows. The first banks to exit the fossil fuel debt market have little impact on
debt flows, as their capital is substituted by other banks. However, a sudden transition is
observed after a certain number of banks have exited the sector, at which point debt flows
sharply contract. The tipping point depends critically on the stringency of regulations; the
number of banks that must exit the sector before the tipping point is reached decreases rapidly
as regulatory rules are tightened. Moreover, the tipping point is reached sooner if large banks
(G-SIBs) move first and coordinate their actions.
Suitable macroprudential regulation, such as capital requirements rules, or other policy
measures which cap a banks' fossil fuel assets, will deliver a managed decline in fossil fuel
lending. On the one hand, overly stringent requirements could precipitate a tipping point too

---

[3] Macroprudential policy is composed of different tools having the goal of preserving financial
stability. This includes making the financial system more resilient to losses and limiting the
build-up of vulnerabilities in order to mitigate systemic risk and ensure that financial services
continue to be provided effectively to the economy.

early, leading to a disruptive transition in which the failure of fossil fuel companies is too
widespread to be managed sustainably. On the other hand, loose requirements resulting in a
late, or non-existent, tipping point could delay the emissions reductions necessary to keep
Paris temperature targets within reach. Such rules can be developed by formal standard-
setting bodies and prudential regulators such as the Basel Committee on Banking Supervision
and the Financial Stability Board. At the same time, banks could strategically coordinate their
transition plans to increase their collective impact on debt markets through voluntary alliances
such as the Net Zero Banking Alliance (NZBA 2021), to which many of the most influential
banks in the sector are signatories. Here again, we see the articulation of the two basic
mechanisms activatable within the financial system: market-driven risk/return dis/incentives,
and purpose-driven alignment strategies.
Finally, the utilisation of policy mixes that incorporate a combination of command-and-control
and market-based instruments can be likened to 'kicks' that yield positive outcomes for the
transition to a net-zero carbon economy. These could take the form of policy mandates such
as progressive emissions reduction targets, environmental and industrial regulation,
mandated transition planning, green central banking, green infrastructure requirements and
building codes working alongside market-shifting initiatives like carbon pricing, climate-related
financial disclosures, green subsidies, risk underwriting mechanisms and green certificates.
Recent advancements in modelling have demonstrated that these policy combinations have
the potential to initiate a virtuous cycle, driving technological development, reducing the overall
need for public investment, and simultaneously stimulating employment and economic growth
(Wieners et al. 2023; Lamperti et al. 2020; Lamperti and Roventini 2022; Stern and Stiglitz
2023). Moreover, such positive feedback loops significantly lessen the reliance on carbon
taxes by decreasing their intensity. As a result, this enhances their political acceptability and
potentially triggers another tipping point.
The importance of these tipping points in the financial system will ultimately be defined by the
impact they have on the decarbonisation of different sectors in the economy. A regulatory
mandate or a market-based measure that affects only a subset of the financial market, such
as commercial banks or publicly-listed companies, or only impacts flows from  a specific
country or geography (such as EU-wide), could potentially  lead to redistribution of high-carbon
assets across the financial system rather than their absolute reduction,  resulting in limited
economy-wide decarbonisation. Broad-based policies are thus needed to influence a sizable
portion of markets to pass a tipping point where financial markets are unable to adequately
substitute the money leaving high-carbon assets. Further, different financial policies are likely
to draw a diverse response from market participants, such as the impact of capital reserve
requirements for the banking sector or carbon disclosure requirements for asset managers
that might not have a significant impact on other actors like private equity funds. Thus a
combination of financial policies will be needed to cover the various investment channels in
the financial system. Specific policies will also be needed to spur investments in climate
projects by mandating investments in specific green sectors or providing market-based
incentives that influence the risk-adjusted returns of these projects. This will ensure that capital
flows diverting from high-carbon sectors reach their intended target and lead to
decarbonisation of the economy. Similarly, targeted international flows to developing countries
will result in an expansion of green sectors in these countries and thereby sustainable
development. Further, the interlinkage between financial and other economic systems needs
to be acknowledged. Policy mixes work well because they influence multiple systems and
attempt to gain non-linear benefits through reinforcing mechanisms and positive tipping points.
**4. Conclusion**
As of today, the financial sector is contributing to a projected +3°C global warming scenario
by 2100. The financial system itself does not inherently favour any particular climate objectives
ex ante. To successfully shift the economy towards a net-zero emission path, it becomes
crucial to harness the potential of tipping points in the financial system in order to contribute
to this transition in its full capacity, by enabling and accelerating the necessary capital
reallocation. These elements can play a pivotal role in redirecting economic activities towards
sustainable practices.
Taken together, the mechanisms detailed above highlight examples of the system-wide tipping
points' potential within sustainable finance and emphasise the necessity of policy interventions
to activate and capitalise on these dynamics. Through the alignment of expectations,
promotion of herding behaviour, utilisation of public finance, reduction of capital costs and
attainment of low-carbon investment thresholds in developing economies, and implementation
of robust financial regulation and policies, the financial system can assume a central role in
expediting the shift towards a net-zero carbon economy.
Regulation plays a critical role in driving tipping points within the financial sector and its role
has become increasingly prominent in recent years. A climate risk information ecosystem has
evolved with standards for climate-related financial data, assessing climate risk impacts,
transparency requirements, green taxonomies, green labels for financial products and
transition risk management plans (International Monetary Fund 2023). Robust monitoring and
supervision by entities like central banks and financial regulators are forcing financial
institutions to move faster and more decisively than market signals alone would prompt them
to do. In this regard, policy makers and financial authorities hold the potential to take a leading
role in steering the financial system towards transformative tipping points, dedicated to
financing the transition to a net-zero carbon economy. A just transition needs investments in
all parts of the economy and society. This will in-turn require policy combinations incorporating
both market-based and structural change instruments to work effectively to deliver
opportunities and investment-friendly conditions while avoiding trade-offs between prudential
behaviour and a shift in asset allocation by financial institutions to low-carbon activities. As
key stakeholders increase their efforts to guide the financial system, leveraging all the
available tools and exploring new avenues, they can also create a coordinated momentum
with industrial policy makers. In this way, financial and economic policies can be more
effectively aligned to support sustainable industries and practices. This collaboration further
strengthens the potential to tip the financial system into a new momentum, where the
identification of critical intervention points can lead to the amplification of sustainable
investments, mitigate risks, and foster transformative changes in the practices of the financial
sector.

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
