# Peer review of "Driving sustainability transitions through financial tipping points"

_EGUsphere, 2023_

## Referee Comment (RC1)

Charlotte Gardes-Landolfini
Climate change, energy, and financial stability expert – IMF
CGardes@imf.org

Thank you for the authors for the opportunity to review this paper. My comments relate to **(i)** the framing of the paper and the climate impact of policy measures, and **(ii)** technical/secondary comments on some key concepts and/or initiatives laid out in the paper.

*Framing and climate impact*

o **The paper would benefit from greater critical thinking on the pass-through between financial sector policies (e.g., disclosure requirements, accounting requirements, prudential regulation (pillar I), etc.) and real economy decarbonization**. The risk of a mere redistribution of GHG emissions between financial market participants is real and explaining better to what extent any kind of financial sector policy (acting on return and/or risk) would have an impact on real-world decarbonization would be essential to the paper. Depending on the type of economy, industry, and financial market participant, as well as on the type of policy, these links are not evident and would need to be substantiated (even from a theoretical standpoint).

o **The paper would benefit from clarifying the application of tipping points from an alignment standpoint (i.e., financing the low-carbon transition) and a climate risk standpoint.** The type of tipping point (and associated policy (prudential and/or disclosure oriented, for instance)) may differ depending on the objective to be achieved. Yet, there may be some trade-offs between risk management and transition financing, that should be acknowledged, and could help refining the paper's conclusions on regulation.

o **Could the paper elaborate on the link between climate-related disclosures and repricing?** The paper appears to mix financial and extra-financial disclosures and/or accounting, and to make a direct connection between transparency and repricing of assets. Yet, for a variety of political economy, incentive-related, and operational reasons, this connection remains to be proven – despite increasing regulatory, policy and/or voluntary initiatives in this area. The focus on "informational policy" to drive change has been criticized by the literature and should not be overlooked.

*Technical/secondary comments*

o **The concept of tipping points is used in an innovative manner in this paper and would benefit from clarification.** While this concept is usually contemplated from a climate science standpoint, its application to climate/sustainable finance policy is interesting. Yet, it would be important to connect it in this paper to the climate science domain and explain how similarly and/or differently it would need to be tackled from a policy perspective.

o **A synthesis of the main tipping points at stake, their transmission channels, and their potential impact, would be extremely relevant**. The paper is of a theoretical nature and lays out a key concept for future research and policy design. Yet, it may remain unclear for the reader to what extent these tipping points are linked (or not) to each other. A synthesis diagram would be relevant and could gain as much traction as the widespread climate risk/macroeconomy/financial system diagram that has been first designed by the NGFS, for instance.

- o **Potentially relevant messaging could be extracted from the latest Chapter of the IMF Global Financial Stability Report (October 2023), on a) the policy mix to be achieved between carbon pricing and non-pricing/structural policies (including the climate information architecture) and b) the allocation of G-SIB lending to fossil fuel companies.** Greater detail may be found here: Global Financial Stability Report, October 2023 (imf.org).

- o **The paper would benefit from:**
  - A clarification of the low-carbon nature of the transition – currently defined as zero and negative emission assets. The focus on negative emissions may be controversial. The need to finance emission *reductions* in carbon-intensive assets may also be raised.
  - The reference to "carbon" emissions. While this is a commonly used terminology in academic papers, the term "greenhouse gas" should be preferred. Non-CO2 emissions are also extremely important (e.g., methane) and should not be overlooked in policy thinking.
  - The different nature of GFANZ (private-led, voluntary) and NGFS (central bank-led, voluntary) should be acknowledged. Both initiatives are very different from a political economy perspective, but also in terms of deliverables, ambition, members, and institutional dynamics. The paper could clarify what is meant by "NGFS opened a new governance framework to better coordinate and regulate the role of finance in addressing climate change".
  - The difference between climate objectives (and what are they in the paper's framing?) and sustainability objectives. The paper uses both terms and concepts alternatively, despite their critical differences.

---

## Author Comment (AC1)

**Driving sustainability transitions through financial tipping points**
Nadia Ameli, Hugues Chenet, Max Falkenberg, Sumit Kothari, Jamie
Rickman, and Francesco Lamperti
https://doi.org/10.5194/egusphere-2023-1750

**Reply to reviewers – R1**

Here we provide our point-to-point replies to the comments raised by reviewer 1.
To better highlight the replies, we write them in italic dark blue. As we are not supposed
to send over the revised manuscript at this stage, our replies are written using the future
tense. Before moving to the replies, we take the occasion to thank the reviewer,
Charlotte Gardes-Landolfini, for her constructive comments.
* * *
**RC1**: 'Comment on egusphere-2023-1750', Charlotte Gardes, 03 Oct 2023
Find attached my comments on this fantastic paper. I would be happy to discuss them
with the authors, if needed,

All the best,

Charlotte Gardes-Landolfini (IMF)

Citation: https://doi.org/10.5194/egusphere-2023-1750-RC1

Charlotte Gardes-Landolfini
Climate change, energy, and financial stability expert – IMF CGardes@imf.org

Thank you for the authors for the opportunity to review this paper. My comments relate
to **(i)** the framing of the paper and the climate impact of policy measures, and **(ii)**
technical/secondary comments on some key concepts and/or initiatives laid out in the
paper.

***Framing and climate impact***

o **The paper would benefit from greater critical thinking on the pass-through
between financial sector policies (e.g., disclosure requirements, accounting
requirements, prudential regulation (pillar I), etc.) and real economy
decarbonization**. The risk of a mere redistribution of GHG emissions between financial
market participants is real and explaining better to what extent any kind of financial
sector policy (acting on return and/or risk) would have an impact on real-world
decarbonization would be essential to the paper. Depending on the type of economy,
industry, and financial market participant, as well as on the type of policy, these links

are not evident and would need to be substantiated (even from a theoretical standpoint).

*We thank the reviewer for this comment, with which we completely align. We decided to amend the paper in two ways, to better discuss the issue pointed out by the reviewer. In the introduction of the paper, we will add a full paragraph to better emphasize the theoretical links we envision between financial sector policies and the decarbonization of the real economy, distinguishing between key sectors and financial actors. We agree the risk of a mere redistribution is real and that the manuscript can benefit from a more critical and detailed framing of the channels through which financial policies may affect emission mitigation. Indeed, we also will also modify the text at page 8 (where we discuss different financial policies and their expected effects on financial flows and emissions dynamics) to spell out the mechanisms at stake in more detail.*

*Finally, we believe some of our replies to the comments below could also help clarify the pass-through between financial policies and the decarbonization of real activities.*

o **The paper would benefit from clarifying the application of tipping points from an alignment standpoint (i.e., financing the low-carbon transition) and a climate risk standpoint.** The type of tipping point (and associated policy (prudential and/or disclosure oriented, for instance)) may differ depending on the objective to be achieved. Yet, there may be some trade-offs between risk management and transition financing, that should be acknowledged, and could help refining the paper's conclusions on regulation.

*Thanks for this remark. We had some issues at interpreting the exact point the reviewer is making on the "type" of tipping point. Indeed, in the text, we do not categorize nor taxonomize tipping points. However, we fully agree with the second part of the comment: trade-offs may well emerge between risk management and transition financing. Hence, we will amend the revised version of the paper to fully acknowledge and further discuss such an aspect. Specifically, we will amend section 2.1 and link the discussion on the trade-off to our treatment of sensitive intervention points. In addition, as suggested by the reviewer, we will revise the paragraph of our conclusions (pag. 9, lines 30-43) concerning regulation, and will better highlight the role of financial institutions and the consequences of using regulatory policy instruments (such as prudential requirements or disclosure obligations), emphasizing the trade-off.*

o **Could the paper elaborate on the link between climate-related disclosures and repricing?** The paper appears to mix financial and extra-financial disclosures and/or accounting, and to make a direct connection between transparency and repricing of assets. Yet, for a variety of political economy, incentive-related, and operational reasons, this connection remains to be proven – despite increasing regulatory, policy and/or voluntary initiatives in this area. The focus on "informational policy" to drive change has been criticized by the literature and should not be overlooked.

*This point is indeed crucial. The perceived shortcut is not intentional and needs to be adjusted. We will do so by clarifying what we intend to mean as the role of disclosure (pag. 5, first paragraph, lines 6-10) and insisting indeed that "transparency is not enough". Incidentally, there are also contributions by the authors that underline such aspects from an empirical and theoretical perspective. [Ameli, N. et al. (2019) 'Climate finance and disclosure for institutional investors: why transparency is not enough', Climatic Change. doi: https://doi.org/10.1007/s10584-019-02542-2.; Lamperti et al. (2021) Three financial policies to address climate risks, Journal of Financial Stability. Doi: https://doi.org/10.1016/j.jfs.2021.100875]*

***Technical/secondary comments***

o **The concept of tipping points is used in an innovative manner in this paper and would benefit from clarification.** While this concept is usually contemplated from a climate science standpoint, its application to climate/sustainable finance policy is interesting. Yet, it would be important to connect it in this paper to the climate science domain and explain how similarly and/or differently it would need to be tackled from a policy perspective.

*Thanks for this remark. Though the concept of "tipping point" is not novel in the economics and financial literature, a clear and shared definition is still difficult to find, especially given the increasing popularity of the concept. We will improve the clarity of the paper by discussing the concept of "tipping point" early in the text (starting from the Introduction) and connecting it to the literature on tipping points and threshold effects both in climate and social science. We will build on the general definition of tipping point by Lenton (2020): "Tipping points describe critical thresholds in complex systems that - if crossed - can lead to qualitatively" and the one of social tipping processes by Tabara et al. (2018) "'Social tipping processes' describe how social, political, economic or technological systems can move rapidly into a new system state or functioning". We highlight that both papers focus on the notion of "positive tipping point", which is close to that examined in this paper. [Lenton, T. M. (2020). Tipping positive change. Philosophical Transactions of the Royal Society B, 375(1794), 20190123.; Tàbara, J. D., Frantzeskaki, N., Hölscher, K., Pedde, S., Kok, K., Lamperti, F., ... & Berry, P. (2018). Positive tipping points in a rapidly warming world. Current Opinion in Environmental Sustainability, 31, 120-129.]*

o **A synthesis of the main tipping points at stake, their transmission channels, and their potential impact, would be extremely relevant**. The paper is of a theoretical nature and lays out a key concept for future research and policy design. Yet, it may remain unclear for the reader to what extent these tipping points are linked (or not) to each other. A synthesis diagram would be relevant and could gain as much traction as the widespread climate risk/macroeconomy/financial system diagram that has been first designed by the NGFS, for instance.

*Very good suggestion. We will endeavour to improve the paper with such graphical summary, linking together the different tipping elements we discuss in the text, including their potential influences. In particular, we will update section 2. wherein we elaborate on the potential for tipping points in the financial sector. However, we will also be very cautious in our discussion of such interactions. Differently than for other "tipping elements" in the coupled climate-economy system, where cascades of tipping points have been already proposed and evaluated, there is still scarce and unconsolidated evidence for positive tipping dynamics in the financial sector; actually, to the best of our knowledge, section 2.2 of the present paper is the most comprehensive attempt at collecting empirical and modelling evidence of such dynamics. Though a full treatment of the linkages between the tipping elements in the financial sector is beyond the scope of the paper, we fully agree that a more synthetic and better designed wrap up of the tipping points at stake would be extremely relevant.*

o **Potentially relevant messaging could be extracted from the latest Chapter of the IMF Global Financial Stability Report (October 2023), on a) the policy mix to be achieved between carbon pricing and non-pricing/structural policies (including the climate information architecture) and b) the allocation of G-SIB lending to fossil fuel companies.** Greater detail may be found here: Global Financial Stability Report, October 2023 (imf.org).

*This suggestion is appealing, and we thank the reviewer for pointing out the reference. We will update section 2.1 (pag.5), section 2.2 (pag. 8 and pag. 9) and the conclusions (pag. 9) to improve our discussion of the role of policy mixes in potentially inducing positive tipping points, balancing between "princing" and "non-pricing" policies. Further, we will also connect to the IEA (2021) Net-zero report, which also hints to the efficacy of policy combinations including both type of policy instruments. With respect to the role of G-SIB lending to fossil fuel companies, we will amend section 2.1 (pag. 5) and emphasize the potential for targeted policies at managing risks and inducing portfolio rebalancing dynamics, which indeed connects to our discussion of the sensitive intervention points identified by Farmer and colleagues (Farmer et al. 2019, Science; cited in the text) at lines 15-26.*

o **The paper would benefit from:**

- - A clarification of the low-carbon nature of the transition – currently defined as zero and

    negative emission assets. The focus on negative emissions may be controversial. The need

    to finance emission *reductions* in carbon-intensive assets may also be raised.

*Thanks for this suggestion. In general, we believe our focus on negative emissions to be very mild. Negative emissions are only mentioned at line 2 of pag. 1 to stress the concept that reaching net zero with the current and envisioned set of technologies implies some removal of $CO_2$ from the atmosphere in additional to natural sinks. We do not think this is controversial, especially given the recent report of IPCC WGIII on*

*mitigation and the available set of 1.5 scenarios. Differently, we agree with the reviewer that the need to finance emission reduction in carbon-intensive assets should also be mentioned. We will amend the introduction (pag. 1) to better emphasize such an aspect.*

- - The reference to "carbon" emissions. While this is a commonly used terminology in academic

  papers, the term "greenhouse gas" should be preferred. Non-CO2 emissions are also

  extremely important (e.g., methane) and should not be overlooked in policy thinking.

*Fully agreed. We will change the terminology employed in the text.*

- - The different nature of GFANZ (private-led, voluntary) and NGFS (central bank-led,

  voluntary) should be acknowledged. Both initiatives are very different from a political economy perspective, but also in terms of deliverables, ambition, members, and institutional dynamics. The paper could clarify what is meant by "NGFS opened a new governance framework to better coordinate and regulate the role of finance in addressing climate change".

*Fully agreed. These two different frameworks need better articulation. Nevertheless, they also share common logics, in that both organisations' membership is voluntary and none are policy prescriptive. We will clarify this sentence, explaining that the NGFS was key and the first such initiative to instrumentally foster implicit coordination between financial (mostly delegated) authorities at global level, having in mind that their top-of-the-pyramid position in the financial system has great signalling and regulating power to infuse the whole panel of financial institutions worldwide.*

- - The difference between climate objectives (and what are they in the paper's framing?) and sustainability objectives. The paper uses both terms and concepts alternatively, despite their critical differences.

*We will clarify this issue. The initial idea was to set the reflexion on climate change (both tipping and Paris objectives) and to expand it similarly to other tipping mechanisms and sustainability issues (starting with biodiversity / Kunming-Montreal objectives, which also use finance as an instrumental means of implementation). However, given the focus of the paper and the available space, we decided to stick to climate related objectives, which are translated in global warming's stabilization between 1.5 and 2 degrees, aligning to the Paris agreement goals. We will amend the introduction (and the rest of the text, where needed) to clarify the present aspect.*

---

## Author Comment (AC2)

**Driving sustainability transitions through financial tipping points**
Nadia Ameli, Hugues Chenet, Max Falkenberg, Sumit Kothari, Jamie
Rickman, and Francesco Lamperti
https://doi.org/10.5194/egusphere-2023-1750

**Reply to reviewer – R2**

Here we provide our point-to-point replies to the comments raised by reviewer 2.
To better highlight the replies, we write them in italic dark blue. As we are not supposed to send over the revised manuscript at this stage, our replies are written using the future tense. Before moving to the replies, we take the occasion to thank the anonymous reviewer for her/his constructive comments.

--

**RC2**: 'Comment on egusphere-2023-1750', Anonymous Referee #2, 25 Dec 2023

"Financial tipping points" is an interesting subject in the interdisciplinary research on critical thresholds in the climate system, and also in the fast-evolving field of climate finance. The main focus of this paper is on "positive" tipping points, in particular on how "the financial system can assume a central role in re-orienting economies onto a net-zero course".

The structure is reasonably clear. After the introduction, there is a listing of concepts on the "potential for tipping points" in finance, followed by a run-through of "empirical and modelling evidence", and some conclusions. The narratives of the 5-6 potential financial tipping points are summarized well. Readers with a certain understanding of the climate finance "jargon" can certainly learn from that.

*We would like to thank the reviewer for this generic assessment of the paper. Given the nature of the journal, which is not a finance-oriented outlet, we will try to revise the text – and especially the introduction and the conclusions – to avoid giving for granted terminology and make the key messages more easily accessible to a generic/climate science oriented audience.*

It is not easy to assess the scientific contribution of this summary paper. For a literature survey, it would not be critical and wide-ranging enough. It feels more like an introduction to a research agenda, with various references to scholarly work undertaken elsewhere. There is not much time spent on the definition and critical examination of concepts and insights. There are hardly any figures, stats, tables or graphs – quite striking for a finance paper. A few examples are given but it is often difficult to distinguish "positive" and "normative" statements, exemplified by the fact that the word "can", e.g., is used 44 times.

*We fully understand the concerns of the reviewer. This paper is part of a special issue devoted to the analysis of tipping points across a large number of subjects and disciplines. The aim of the special issue is to consolidate the knowledge on both "good"*

*and "bad" tipping points that are relevant for climate change, taken at large. For what concerns mitigation, the focus is mostly on updating about positive tipping elements leading to transformative change. The full description is available here:* https://esd.copernicus.org/articles/special_issue1247.html

*As correctly understood by the reviewer, this paper should be seen as an attempt to provide an introduction to a novel transdisciplinary research agenda studying tipping dynamics leading to "positive" transformations of the financial sector. In doing so we provide a critical overview of the mechanisms that may drive to such changes and survey the (limited) available modelling and empirical evidence in the field. Further, we believe our paper is definitely more "positive" than "normative": we attempt at describing mechanisms and dynamics of our system, without specifying how the system itself should be organized or modified. Sometimes we speculate on the possible effects of some policies or changes in the existing conditions, trying to back such effects with the available evidence.*

*We hope clarifying these background and the paper's goal can help the reviewer in her/his assessment. Further, we will modify the introduction (pag. 1) to better position our paper and specify its main goals/objectives; the same will be done for conclusions.*

For my personal taste, a generic approach using the "financial system" is probably too broad a brush. Sustainable finance and impact investing have made considerable progress in recent years, and so has regulation in the field. New facts are being established, although unevenly across sectors, geographies, financial and political institutions. Academic research needs to be careful not to fall too far behind the "factual" curve.

*We hope that the reply to the previous point also helps here. Though we recognize an enormous progress in the fields of sustainable finance, climate finance and impact investing across a number of specific issues (from asset stranding to ESG investments, all the way to carbon risk pricing and more), the focus of the present paper is introducing to (positive) tipping points within the financial sector. We chose to adopt a "generic" system perspective on purpose, to present and discuss a series of features and mechanisms that are good candidates to generate tipping and self-reinforcing dynamics across multiple markets and sectors of the financial systems. We believe the example of expectation alignment is a good one and may help here. As long as we agree that much of the transition will be induced by policies, aligning expectations by reducing policy uncertainty in the short and long run, has potential to catalyze financial flows, spurring physical investments and R&D, which will increase the expected return of green technologies, thereby generating a virtuous cycle. This can be relevant across different industries (e.g. renewables, batteries, electric vehicles) and types of investors (e.g. public investment banks, private investment banks, pension funds) operating through different markets. Such feedback chain builds on consolidated empirical evidence coming from other sectors and policies (e.g. Gulen and Ion, 2016; Wen et al. 2022), but has potential to apply to climate policy and many related investment channels.*

*We will try to reinforce the description of the mechanisms we envision with robust empirical evidence, but we would strongly prefer to keep our "system-oriented" focus. In addition, we believe the literature is not consolidated enough to clearly identify and discuss sector and geography specific tipping dynamics.*

*[Gulen, H., & Ion, M. (2016). Policy uncertainty and corporate investment. The Review of financial studies, 29(3), 523-564.; Wen, H., Lee, C. C., & Zhou, F. (2022). How does fiscal policy uncertainty affect corporate innovation investment? Evidence from China's new energy industry. Energy Economics, 105, 105767.]*

In terms of minor comments:

- Some more definitions could be given for the non-expert readership.

*We definitely agree with this comment. We will include more definitions (e.g. tipping point, sustainable/climate finance, asset stranding, prudential policy, transformative change etc).*

- Careful with generic statements and consistency. For example, is the financial system "neutral" (page 9) or "conservative" (page 3)?

We apology for the confusion. In the conclusions, at pag. 9, we meant that the financial system – intended as a pool of financial actors interacting through contracts and markets – is not guided by any specific (nor shared) climate objective when conducting its operative activity, especially in absence of policies. *In this sense, it is "neutral" to climate-related goals. We will adjust the sentence accordingly and remove the term "neutral".*

- There is still a diligent job to be done at reconciling references in the text and the reading list.

*Thanks for signaling this issue. We will amend the references and citations and remove the inconsistencies.*

**Citation**: https://doi.org/10.5194/egusphere-2023-1750-RC2

---

## Author Response (AR2)

Response to reviewers:

**Reviewer 1**

I think the paper is an important paper for laying out the research agenda on how the financial sector could be an important transformative tool for greening our economies. However, before publication I would see the following aspects to be addressed:

1. Introduction: The introduction currently presents a rather negative view of finance. It would benefit from highlighting some of the progress made in sustainable finance over recent years, including advancements in regulation and increased volumes. While acknowledging that these efforts are still insufficient, the paper should recognize these developments, as suggested in the first review and point out more clearly where the actual gap lies. Additionally, incorporating insights from the recent World Bank Finance and Prosperity Report Chapter 3, which discusses climate finance instruments used by central banks, regulators, and supervisors, would be beneficial. In this context, the paper could even more clearly identify the existing gaps that hinder positive tipping points in the financial sector.

*Thank you for your insightful comments. We have now addressed these points by expanding the introduction to highlight recent progress in sustainable finance and new tools and instruments from Chapter 3 of the recent World Bank Finance and Prosperity Report. While we acknowledge that these efforts remain insufficient, the revised introduction recognizes these developments and more clearly identifies the persistent gaps.*

2. Clarifying the Tipping Point: The concept of the actual tipping point is not yet clear. The current description suggests a non-linear growth in sustainable finance, but it does not fully explain how sustainable finance will become the new default, even if inducing policies are phased out, which in my understanding would represent a true tipping point rather than just scale.

*Thank you for your insightful feedback on clarifying the concept of a tipping point in sustainable finance. In response, we have refined the discussion to emphasize that a true tipping point represents a self-sustaining transformation in which sustainable finance becomes the standard within financial markets, independent of continued policy support. This clarifies that the tipping point involves structural permanence in sustainable finance, rather than merely an increase in scale.*

3. Linking Financial Sector Policies and Decarbonization: The previous review requested a stronger connection between financial sector policies and the decarbonization of the real economy. The paper should address in more detail the actual transmission channels, e.g. if we are talking only about reductions in basis points or more substantial impacts that finance could have. My point is that if finance is seen as such a transformative tool for greening our economies, then it would be good to be more explicit on how it could do so.

*Thank you for your suggestion to further specify the link between financial policies and decarbonization in the real economy. In response, we have enhanced our discussion to describe the transmission channels more explicitly. We highlight how financial sector policies, through mechanisms like carbon-adjusted capital requirements, influence both capital costs and allocation patterns, creating structural incentives for a shift toward low-carbon investments. This approach clarifies the substantial impact that finance, beyond minor cost reductions, can have on real-economy decarbonization.*

4. Demand for Financing in the Real Economy: The financial sector requires demand for its financing, which is often lacking in the real economy. The paper could explore further where the incentives are and what real-sector policies are needed to complement financial sector efforts. The potential risk of a green bubble may also require further attention in this context.

*Thank you for your insightful feedback regarding the demand for sustainable financing in the real economy and the potential risk of a green bubble. In response, we have expanded our discussion to clarify how real-sector policies can drive demand for sustainable finance and thereby complement financial sector efforts. Specifically, we discuss how instruments such as carbon pricing, renewable energy subsidies, and sector-specific decarbonization mandates create concrete incentives for firms to pursue sustainable financing. These policies anchor green investments in real economic activities, ensuring that capital flows support genuine decarbonization efforts rather than speculative growth.*

*We also address the potential for a green bubble by highlighting the importance of aligning real-sector policies with financial initiatives. By fostering substantive demand for sustainable finance within the real economy, such policy alignment minimizes the risk of asset inflation in green sectors. This integration helps prevent speculative distortions and supports a stable growth environment for sustainable finance, promoting resilience in capital flows. We believe these additions clarify the mechanisms by which real-sector policies can support a balanced expansion of low-carbon investments, reducing the risk of volatility and speculative bubbles.*

5. Figure 1 and High-Carbon Investments: Figure 1 was added in response to previous reviewer comments to conceptualize the positive feedback loops leading to tipping point dynamics. However, it does not adequately address the reduction in high-carbon investments. The paper should discuss regulatory tools to reduce high-carbon investments earlier, leading up to Figure 1.

*Thank you for your suggestion regarding the inclusion of regulatory tools to reduce high-carbon investments. We have addressed this point later in the paper, specifically on page 13, where we discuss the role of macroprudential regulation in reducing fossil fuel lending. In this section, we examine tools such as carbon-adjusted capital requirements and limits on fossil fuel asset holdings, which are designed to systematically decrease capital flows to high-emission sectors shifting flows towards alternative assets. These regulatory measures aim to align financial practices with decarbonization goals and reduce the financial viability of high-carbon investments. We hope this section provides the clarity and depth you were looking for on the use of regulatory tools to support a low-carbon transition.*

6. Trade-offs in Sustainable Finance: The paper indicates that the focus on climate finance could be applied to other aspects of sustainable finance. However, it should also point out the trade-offs between larger climate investments and other sustainability goals, such as nature preservation. Mining is an example where these trade-offs are evident.

*Thank you for your thoughtful feedback regarding the potential trade-offs between climate finance and other sustainability goals, such as biodiversity conservation. We agree that this is an important matter that was not sufficiently highlighted, and added an explanation as well as a specific reference on the issue of biodiversity and climate interactions as seen from finance.*

8. Contradiction on Page 8: The second full paragraph on page 8 seems contradictory. It states that disclosure standards and transition plans could help the transition in an efficient market system but also mentions inherent limits. The messaging here should be strengthened for clarity.

*Thank you for your feedback regarding the clarity of the paragraph on disclosure standards and transition plans. We have revised this section to improve the flow and strengthen the messaging. The updated text clarifies that, while disclosure standards and transition plans can facilitate the transition by supporting efficient market mechanisms, there are inherent limitations to the extent that markets alone can incorporate long-term climate risks due to the uncertainties and complexities involved in the low-carbon transition. We have highlighted that increasingly interventionist regulatory measures are*

*being proposed, particularly in Europe, to help address these limitations and reinforce the impact of disclosures. We hope this revision addresses your concerns and enhances the overall clarity of our argument.*

9. Consistency in Terminology: The paper should ensure consistent use of terms such as GHG versus carbon emissions, which is not always the case.

*Thank you for pointing out the need for consistent terminology. We have reviewed the manuscript to ensure uniform use of terms such as "GHG emissions" and "carbon emissions." Where appropriate, we standardized the terminology to "GHG emissions" for consistency throughout the paper. We appreciate your attention to detail, which has helped us improve the clarity and precision of our language.*